# CareGraph: A Graph-based Recommender System for Diabetes Self-Care Management

## Abstract

In this work, we build a knowledge graph that captures key attributes of content and notifications in a digital health platform for diabetes management. We propose a Deep Neural Network-based recommender that uses the knowledge graph embeddings to recommend health nudges for maximizing engagement by combating the cold-start and sparsity problems. We use a leave-one-out approach to evaluate the model. We compare the proposed model performance with a text similarity and Deep-and-Cross Network-based approach as the baseline. The overall improvement in Click-Through-Rate prediction AUC for the Knowledge-Graph-based model was 11%. We also observe that our model improved the average AUC by 5 % in cold-start situations.

## 1 Introduction

The recent global pandemic has brought with it a permanent shift away from traditional in-person health consultations, towards large digital telehealth platforms that support remote consults. This research is performed in the context of one such platform that provides a mobile application to help people manage chronic diseases such as diabetes, hypertension and obesity. As solution designers, some of the key problems we face is to get our users' attention, foster awareness and encourage them to take actions that help manage their chronic health conditions. Sending notifications that nudge users is one efficient way to encourage engagement. However, user preferences for engaging with these nudges can vary greatly, different users require different persuasion techniques. Even within a homogeneous messaging context, different tones in messages can appeal to different users.

Recommendation systems can be applied to model preference patterns and predict effective personalized nudge notifications. Collaborative recommender systems are well established tools for predicting user preference and have been shown to perform very well, provided there is sufficient information available to model the users preferences, as highlighted by seminal papers in the field such as Resnick et al. (1994). For new or esoteric items, or users, there is frequently a lack of sufficient information to make a good prediction. These conditions, known in the literature as 'cold start' are the focus of our experiments in this paper. Specifically in the context of recommending health nudges, which are created by experts who inherently follow guidelines set by associations such as American Diabetes Association ame (2021). Our primary research question asks whether a knowledge graph can be applied to mitigate cold start problems, in the specific task of recommending a finite set of highly structured health nudge messages.

### 1.1 Knowledge Graph-based Recommendation

Online content and services have undergone a huge volume of growth in recent years. Accordingly, recommender systems are increasingly relied upon to help people get to the right information at the right time and also in the right way [Ricci et al. (2011)]. They help customers shorten their times exploring products or services in various applications such as news portals [Wang et al. (2018b)], E-commerce [Zhang & Jiao (2007), Hwangbo et al. (2018)], accommodations [Haldar et al. (2019)], or music recommendation [Van Den Oord et al. (2013)]. However, our problems are different from these recommendation applications. First, in other applications like E-commerce, users can see a list of products and select their interested items. They can visit websites to search for items of interests anytime they like. In our nudge notifications, our users can just see one nudge at a time, and hence we cannot flush a lot of nudges as it will disturb our users and can negatively impact their level of engagement. Hence, recommending the nudges that match user preferences while also ensuring that there is a diversity in the actions we are recommending, and achieving good performance in as few notifications as possible is a crucial requirement in this setting. Also, we frequently need to

support the creation of new nudges. Within this setting, using embeddings from supervised learning methodologies may not be able to solve our problems. To learn efficient embeddings from those approaches, we require a lot of data for each item. Organizing and aggregating nudge attributes such as themes, tones or objectives can help the cold-start problem as we can transfer the knowledge of user preferences on nudge attributes to new nudges and help us understand and optimize to user behaviors quickly. For example, if a user has positive responses to encouragement-tone nudges, we can send more encouragement-tone based messages to this user.

Knowledge graphs (KG) have been applied in various tasks such as search engines [Yang et al. (2017)], text classification [Hu et al. (2021)] and word embedding [Celikyilmaz et al. (2015)]. They have also been previously introduced to recommendation systems to improve the precision of recommended items and the explainability [Wang et al. (2019)]. A knowledge graph is a directed heterogeneous graph representing real-world objects and relationship among these objects. Nodes in a KG represent entities, which can be items or the attributes that describe the item. The edges represent relations between each entity. Knowledge graphs give recommendation systems the benefit of enriching the semantic relationships among items [Wang et al. (2019)]. When we know users' interests in a KG graph, we can extend the diversity of recommended items from the node connections present in the graph.

Our problem domain is selecting nudge notifications for our mobile applications that are personalized to individual users preferences. We have different types of nudge notifications such as alerting users in regards to their health condition, reminding users to follow their health routine, providing some useful education related to their health, or introducing new services and events to users. We want to personalize nudges to each user to increase their engagement and reduce over-sending nudges that might disturb the users. The contributions of this work are summarized as follows:

- We propose a new method using a neural network that combines user attributes with a knowledge graph to overcome the cold-start problem.
- We conduct experiments demonstrating the effectiveness of our new approach, particularly under cold-start conditions.

## 1.2 Digital Health System for Diabetes Self-Care

This work is based on a system that leverages a Blood-Glucose (BG) monitoring device, and a Mobile App that is used to provide useful information and suggest actions to users that are relevant and helpful for managing Diabetes. As a part of this system, users are provided with short text messages, about 180 characters long. The text is designed to nudge a user to specific actions that are relevant to managing the condition, such as, healthy recipes ideas, articles that explain and motivate users to add exercise to their routine, reminders to check and monitor BG levels, talking to expert coaches etc.

The short text is accompanied with two buttons, one of which is used to guide the target user to perform a specific action (e.g. read content, schedule a call with a coach), and the other can be used to dismiss the recommendation. In this work we call these recommendations on the Mobile-App, Mobile Nudges (or Nudges for short).

## 2 Related Work

### 2.1 Recommender systems in Digital Health

Over the past decade, a rapidly increasing volume of digital information to support clinical decision making has become available to be leveraged by healthcare recommender systems. Automated health recommender systems have been deployed in various domains. Some applications aim for improving lifestyle to be healthier through diet [Elsweiler et al. (2017), Achananuparp & Weber (2016)] or physical activity recommendations [Dharia et al. (2016)]. One such system, known as Pro-fit [Dharia et al. (2016)] applies a hybrid approach to personalize workout sessions based on a user's contextual data and calendar events. Achananuparp & Weber (2016) proposed a healthier food substitute suggestion system by introducing a food-context matrix and applying Singular Value Decomposition to get the low-dimensional representation of each food item. The similarity between two food items is measured by cosine similarity. Narducci et al. (2015) introduced an HealthNet to personalize doctors and hospitals to users, given the user profile and the health data shared by the community. The iDoctor system in Zhang et al. (2017b) provides doctor recommendations by using sentiment analysis from their rating and reviews and Latent Dirichlet Allocation for user preference and doctor features. The hybrid matrix factorization is applied to predict the doctor rating.

Recommendation systems have also been used for optimizing treatment plan such as drug recommendation [Zhang et al. (2017a), Stark et al. (2017)]. Zhang et al. (2017a) applied the concept of neighborhood-based method in a drug-drug interaction prediction to help reduce unexpected effects from using multiple drugs. A drug recommendation system for Migraine diseases proposed by Stark et al. (2017) using a graph database and collaborative filtering approach.

## 2.2 KNOWLEDGE GRAPH EMBEDDING

Knowledge graph embedding (KGE) is a process to transform a knowledge graph into low-dimensional continuous vector space which still preserves the network structure information. The knowledge graph can be represented as a group of triplets, each of which contains two nodes (items or attributes) and the relationship between these nodes. With these triples KGE will project all entities and relations into a low dimensional vector space that preserves their graph structure in these vector representations. There are several approaches for building these representations in the KGE setting. Translation distance models such as TransE [Bordes et al. (2013)], TransH [Wang et al. (2014)], TransR [Lin et al. (2015)] and semantic matching models such as DistMult [Yang et al. (2014)] are some of the popular methods.

## 2.3 KNOWLEDGE GRAPH EMBEDDING IN RECOMMENDATION SYSTEMS

Knowledge graphs have been applied to many recommendation applications. Wang et al. (2018a) proposed the RippleNet, an end-to-end recommendation framework that incorporates knowledge graph into recommendation systems. RippleNet extends user interest of items through links in the knowledge graph which help increasing the diversity. Wang et al. (2018b) introduced Deep Knowledge-aware Network (DKN) for news recommendation. The key component of DKN is knowledge-aware convolutional neural network (KCNN) for incorporating word-level and knowledge-level representations. Zhang et al. (2018) extends a collaborative filtering framework to learn over the knowledge graph embedding. A user-item graph is used, where each connection between nodes depicts how a user interacted with an item, for example $(buy, also\_bought, also\_view)$

Another method for using a knowledge graph in a recommenation system is path-based approach, which uses the connection patterns in the knowledge graph to generate the recommendation. Hete-CF [Luo et al. (2014)], a collaborative filtering recommendation method on Heterogeneous Social Network, which combines different types of meta-paths (user-user, user-item and item-item) and calculates the similarity. Yu et al. (2014) introduced HeteRec which represents the connectivity between users and items with meta path-based latent features. HeteRec recommendation models are further defined into two levels: global (HeteRec-g) and personalized (HeteRec-p) levels. In the HeteRec-p, users are clustered based on their interests and preferences into subgroups and then learn recommendation models for each user subgroup.

## 3 PROBLEM FORMULATION

Our nudge recommendation can be seen as a binary classification problem where we try to predict the probability that a user will have a positive engagement with a nudge (i.e. accepted the action or suggestion provided by the nudge). Let $U = \{u_1, u_2, ...\}$ and $V = \{v_1, v_2, ...\}$ denote the sets of users and nudges respectively. A user-item interactive matrix is defined as $Y = \{y_{u,v} | u \in U, v \in V\}$ where

$$y_{u,v} = \begin{cases} 1 & \text{if the user } u \text{ positively engaged with the nudge } v \\ 0 & \text{otherwise} \end{cases} \tag{1}$$

We want our model to learn and predict a probability of $u_i$ will click on the nudge $v_j$ or $\hat{y}_{u_i,v_j} = f(u_i, v_j)$.

## 4 METHODOLOGY

Our approach to nudge recommendation is based on a combination of item and user-based collaborative filtering. In our model, a user is represented as a triple $u \in d, i, a$ where $d$ is the a set of demographic profile information such as age and gender; $i$ is the set of clicks on nudges, and $a$ is the set of attributes of the nudges that were effective in engaging that user. The objective is to transform users and items (in our case, nudges) into a vector space and make a prediction based on similarity in the latent space. To learn an embedding of a user profile, we rely on their click history. This information helps us to passively infer preferences based on how they interacted with nudges in the past. If a candidate item

is similar to a user's history of clicked nudges, it is likely that a user will also like this nudge too. However, using only a profile of click history has significant limitations and can result in a narrow set of recommended items, lacking in diversity –essentially a similar narrowness problem to that of a traditional content-based recommender system. To mitigate this problem, which is particularly apparent in cold-start conditions where there are less click data available, we proposed a novel use of "nudge attribute" preferences to user embeddings. Attributes of recommended nudges effectively extend a user's preference profile by creating new edges to explore in the item-to-item graph. Even for a completely new user with no click history, we can infer nudge attribute preferences from the demographic profile until preference data becomes available through click interactions.

The model training occurs in two parts. We first train knowledge graph embeddings to generate nudge, nudge attributes and relation type embeddings. Next, we construct a deep neural network model to predict the click-through rate (CTR) probability of a user $u_i$ on a nudge $v_j$. The user embeddings $e_{u_i}$ (section 4.2.1) are constructed by combining the user click history embeddings $h_{u_i}$, nudge attribute preference embeddings $n_{u_i}$ and user profile embeddings $p_{u_i}$. To learn nudge attribute preference embeddings, we apply the concept of attention mechanism Vaswani et al. (2017) to get a set of values for our attention-based network layer. A nudge prediction is then calculated from a function of $\boldsymbol{e}_{u_i}$ and $\boldsymbol{e}_{v_j}$ where $e_{v_j}$ is an embedding of nudge $v_j$.

## 4.1 KNOWLEDGE GRAPH FOR DIABETES SELF-CARE

We evaluate our proposed models for CTR prediction on our mobile nudge dataset, which we described in Section 1.2. The actions suggested to the users in the nudges cover themes such as monitoring their blood glucose level, improving physical activity, adhering to medication etc. These actions may have different levels of effort to complete. For example, requesting a healthy recipe might be done directly on the mobile application, while talk to a coach or an expert might require the user to find an available time and schedule. We also have nudges that might recommend the same action but use a different tone in the communication text. Furthermore, the nudges can also be categorized based on the intent of support the nudges are be providing, such as providing knowledge or requesting preferences through short questions. We use such categorizations that are used to create new nudges for the users as the attributes for the nudges. These attributes capture not only the theme and intent of the nudge but also some information around the experience accessing the working on the action recommended by nudge.

Table A shows seven attribute types with descriptions and examples values relating to a nudge *"In a dinner rut? Our meal plan is packed with tasty, colorful, and super-healthy recipes, Want it?"*. These attributes and the associated values have been developed by working with the business and subject matter experts.

Figure 3 shows two example health nudges with their associated, and sometimes overlapping attributes. The full set of nudges and attributes in our library are used to define a graph that becomes the initial input to the CareGraph recommender system. The nudges and the attribute values form the nodes of the graph, as highlighted in Figure 3. For example, the blood-glucose monotoring nudge and the holiday nutrition nudge both share the attribute that they send a user to a website for additional information, and that they foster broad awareness of the chronic health condition. More formally, the attribute types are the relations that define the edges of the graph, i.e. the edge describes how a nudge is related to the attribute. This can be represented in a form of triples $(h, r, t)$ where $h$ is the nudge (head), $r$ is the attribute type (relation) and $t$ is the attribute value (tail). For example, if a nudge $CGM\_CONTENT\_BGM\_COMPARE$ has $monitoring$ as the $theme$, then this relationship is represented as $CGM\_CONTENT\_BGM\_COMPARE \xrightarrow{\text{theme}} monitoring$, and forms the triplet $(CGM\_CONTENT\_BGM\_COMPARE,\ theme,\ monitoring)$.

While this is a useful representation for the structure with which nudges are created and described, it will be beneficial to develop continuous representations that can be used for training a recommender model. These representations are called knowledge graph embeddings, and are described in Section 2.2. There are various algorithms that can be used develop these embeddings. Given the relatively simple structure of the graph and the simplicity of the algorithm we use the standard *TransE* algorithm [Bordes et al. (2013)] in our methodology.

## 4.2 CAREGRAPH MODEL

The *CareGraph* model, shown as a flow diagram in Figure 1 is trained in two phases. We first train knowledge graph embeddings for nudges, nudge attributes and attribute types embeddings. Next we

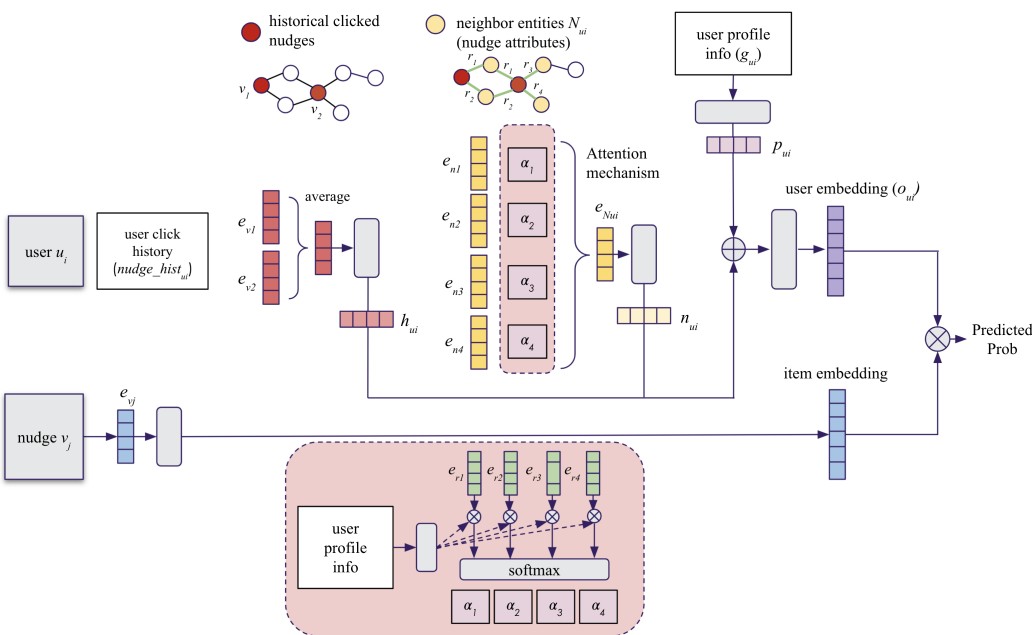

Figure 1: CareGraph model architecture.

construct a deep neural network model to predict the CTR probability for a target user $u_i$ on a nudge $v_j$. The user embeddings $e_{u_i}$ (section 4.2.1) are constructed by combining user click history embeddings $h_{u_i}$, nudge attribute embeddings $n_{u_i}$ and user profile embeddings $p_{u_i}$. The prediction is calculated from a function of $\boldsymbol{e}_{u_i}$ and $\boldsymbol{e}_{v_j}$ where $e_{v_j}$ is an embedding of nudge $v_j$.

### 4.2.1 USER EMBEDDING

**User click history embedding** A user's click history helps us understand which nudges the user liked in the past. We gather all clicked nudges that the user $u_i$ clicked in the past and construct a click history list $nudge\_hist_{u_i}$ where $nudge\_hist_{u_i} = \{v_k | v_k \in V, y_{u_i, v_k} = 1\}$. For users who have never clicked on any nudge, their click history will be an empty list. To represent $nudge\_hist_{u_i}$ as a embedding vector $h_{u_i}$, we use embedding vectors for all $v_k \in nudge\_hist_u$ from our knowledge graph embeddings and use the equation 2 to aggregate them.

$$\boldsymbol{h}_{u_i} = \boldsymbol{W}_m(\frac{\sum_{v_k \in nudge\_hist_{u_i}} \boldsymbol{e}_{v_k}}{\|nudge\_hist_{u_i}\|}) + \boldsymbol{b}_m \tag{2}$$

where $\boldsymbol{e}_{v_k}$ is an embedding vector of the nudge $v_k$ from the knowledge graph embedding. In the equation 2, we first average all embeddings in $nudge\_hist_{u_i}$ and apply a linear function where $\boldsymbol{W}_m$ and $\boldsymbol{b}_m$ are transformation weight and bias respectively. For a user whose click history is empty, the $\boldsymbol{h}_m$ will be a zero vector.

**Nudge attribute embedding** In user click history embeddings, we focus only on nudge embeddings that the user interacted with in the past. We can leverage nudge attribute nodes to extend user preferences by propagating to neighbor entities around users' historical clicked nudges. Since neighbors of nudge entity are nudge attributes entities, we refer to this as the set of *attribute embeddings* $\boldsymbol{n}_{u_i}$.

From a intuitive motivation that different users have different preference on nudge attributes, for example, a user may be interested in a formal clinical information nudge than a cheerful-tone encouragement nudge, we should use user preference weights on nudge attributes instead of averaging them directly as in equation 2. Given a set of nudge attributes and their relation types, we apply the concept of attention mechanism Vaswani et al. (2017) to learn user preference weights of each nudge attribute.

Considering an nudge entity $h$ in the knowledge graph where its connections are represented in the form of $(h, r, t)$, The set of neighbor entities, or nudge attributes, of a user $u_i$ is $N_{u_i} = \{t | h \in nudge\_hist_{u_i}, (h, r, t) \in kg\}$ and the relation types $R_{u_i} = \{r | h \in nudge\_hist_{u_i}, (h, r, t) \in kg\}$. For a user whose click history list is empty, we randomly select a set of nudge attribute entities. The embedding aggregation of nudge attributes can be formulated as

$$e_{N_{u_i}} = \sum_{n_i \in N_{u_i}} \alpha_i e_{n_i} \tag{3}$$

Where $\alpha = (\alpha_1, \alpha_2, ... \alpha_i)$ is an attention weight vector. We calculate the vector $\alpha$ from user profile features $g_{u_i}$ of the user $u$ by the following formula:

$$\alpha_i = \frac{e_{r_i}^T(W_r g_u + b_r)}{\sum_{r_j \in R_{u_i}} e_{r_j}^T(W_r g_u + b_r)} \tag{4}$$

The $W_r$ and $b_r$ a transformation matrix and bias respectively. It will transform the user profile feature $g_u$ into the $r$ vector space and then perform the inner product with the embedding vector $e_{r_i}$ of the attribute type $r_i$. A softmax function is used to compute user relation type preference weights, and then the final nudge attribute embedding $n_u$ is calculated from this equation:

$$n_{u_i} = W_n e_{N_{u_i}} + b_n \tag{5}$$

**User profile embedding**  We also include explicit user features into the user embedding. All user raw data is transformed into a user feature vector and then apply a function to get the final representation.

$$p_{u_i} = W_p g_{u_i} + b_p \tag{6}$$

**Information Aggregation**  The last step to get the final embedding of the user $u_i$ is by combining user history embedding $h_{u_i}$ from equation 2, nudge attribute embedding $n_{u_i}$ from equation 5, and user profile embedding $p_{u_i}$ from equation 6. We concatenate all three embeddings and transform it by a linear function to get the final user embedding $e_{u_i}$.

$$o_{u_i} = h_{u_i} \oplus n_{u_i} \oplus p_{u_i} \tag{7}$$

$$e_u = W_u o_{u_i} + b_u \tag{8}$$

### 4.3 MODEL PREDICTION

To calculate the probability that the user $u_i$ will click on the nudge $v_j$. We use the final user embedding $e_{u_i}$ from equation 8 and nudge embedding $e_{v_j}$ of the nudge $v_j$ from knowledge graph embedding. We predict a user engagement score from the matching score by

$$\hat{y}_{(u_i, v_j)} = \sigma(e_{u_i}^T(W_v e_{v_j} + b_v)) \tag{9}$$

where $\sigma$ is a nonlinear function, similar to the sigmoid function. We can see from the equation 9, the CTR prediction of user $u_i$ and nudge $v_j$, $\hat{y}_{(u_i, v_j)}$, is dependant on how similar the user embedding $e_{u_i}$ is to the nudge embedding $e_{v_j}$.

## 5 CAREGRAPH WITH TEXT EMBEDDING

The CareGraph model supports addition of new features such as text embeddings. A basic nudge is a short text message, and we can aggregate these embeddings into the model to improve its performance. Here the premise is that text similarity across the broader library of nudges can be leveraged to improve the relevance of nudges for a target user. In this section, we introduce two such extensions of the base model architecture: *CareGraph + TextSim* and *CareGraph+TextEmbedding*.

**CareGraph+TextSim**  The approach taken in this model stems from our intuitive assumption that if a text message of a nudge $v_j$ is similar to contents in historical clicked nudges of user $u_i$, it will be likely that a user $u_i$ will also like the nudge $v_j$. With this assumption, we add text similarity score at the prediction step. The other parts of model are still the same. The text similarity score $t_{v_j}$ is be calculated by equation 10

$$t_{v_j} = \frac{\sum_{h_k \in nudge\_hist_{u_i}} dist(x_{h_k}, x_{v_j})}{|nudge\_hist_{u_i}|} \tag{10}$$

where $x_v$ is a text embedding of the nudge $v$ and $dist$ is a distance function of text embedding, such as cosine similarity. The CTR prediction score in equation 9 will be replaced with

$$\hat{y}_{(u_i, v_j)} = \sigma(w_1 e_{u_i}^T e_{v_j} + w_2 t_{v_j}) \tag{11}$$

**CareGraph+TextEmb** Another way to include text embedding into the CareGraph model is to include them as part of the user embedding $e_u$ in equation 7 and 8 and nudge embedding in $e_{v_j}$.

We first get text embeddings of all nudges in $nudge\_hist_{u_i}$ and aggregate them by simple averaging.

$$q_{u_i} = \frac{\sum_{v_k \in nudge\_hist_{u_i}} \boldsymbol{x}_{v_k}}{|nudge\_hist_{u_i}|} \tag{12}$$

$$\boldsymbol{t}_{u_i} = \boldsymbol{W}_t \boldsymbol{q}_{u_t} + \boldsymbol{b}_t \tag{13}$$

To include text embedding $\boldsymbol{t}_{u_i}$ into user embedding $e_u$, we update the equations 7 as follow:

$$\boldsymbol{o}_{u_i} = \boldsymbol{h}_{u_i} \oplus \boldsymbol{n}_{u_i} \oplus \boldsymbol{p}_{u_i} \oplus \boldsymbol{t}_{u_i} \tag{14}$$

and then calculate the final $e_u$ in 8. In the equation 9, we add text embedding of the nudge $v_j$ into $\boldsymbol{e}_{v_j}$ by

$$\hat{y}(u_i, v_j) = \sigma(\boldsymbol{e}_{u_i}^T(\boldsymbol{W}_v(\boldsymbol{e}_{v_j} \oplus x_{v_j}) + \boldsymbol{b}_v)) \tag{15}$$

## 6 Experiment Setup

The data used for this analysis consists of interactions from 283,743 unique users. As mentioned in the description of the Nudges system, the nudge consists on two buttons, one for taking the suggested action and the other for dismissing the Nudge. For the purpose of this analysis a user accepting the recommended nudge is considered a positive rating or a $click$. The data has been collected from 283,743 users over a period of 12 months, and have an average Click-Through-Rate (CTR) of 16.77%.

### 6.1 Procedure

In order to determine the performance of the proposed model two distinct cases are considered. In the first case we evaluate the improvement in predicting the $click$ behavior across all users and all available Nudges in the data. In the second case, the performance of the model is evaluated in Cold-Start cases. Specifically, the case when new Nudges are added is simulated by leaving one nudge out in the training data and determining the performance in predicting the $click$ behavior for that Nudge. This is a typical situation that is encountered in practice where new content and nudges are created and it is important to direct them to most receptive users in a timely fashion. The performance of the proposed model in both cases is compared to a Deep and Cross baseline model, as described below.

#### 6.1.1 Baseline Model for Performance Evaluation

A Deep and Cross Network (DNC), similar to Wang et al. (2021) is used as the baseline model to predict the $click$ on the Nudge. The DCN model starts with an embedding and a stacking layer to transform categorical features into a dense vector. A Cross Network and a Deep Network work in parallel and their output is combined in the last layer. The Cross Network is composed of cross layers which applied feature crossing at each layer. A cross layer is to calculate this following function:

$$\boldsymbol{w}_{l+1} = \boldsymbol{x}_0 \boldsymbol{x}_l^T \boldsymbol{w}_l + \boldsymbol{b}_l + x_l = f(\boldsymbol{x}_l, \boldsymbol{w}_l, \boldsymbol{b}_l) + \boldsymbol{x}_l \tag{16}$$

where $\boldsymbol{x}_l, \boldsymbol{x}_{l+1} \in \mathbb{R}^d$ is the output vector at the layer $l$-th and $l+1$-th layer, respectively. The $x_0$ is the embedding vector from the embedding and stacking layer. By adding more cross layers into the cross network, it increases the polynomial degree of the input $\boldsymbol{x}_0$.

The deep network is a standard fully-connected feed-forward neural network. For each layer, the output of the subsequent layer $l+1$ is calculated by:

$$\boldsymbol{h}_{l+1} = f(\boldsymbol{W}_l \boldsymbol{h}_l + \boldsymbol{b}_l) \tag{17}$$

where $\boldsymbol{W}_l$ is a weight matrix and $\boldsymbol{b}_l$ is the bias of the layer $l$. The final output of the deep network is an embedding vector $\boldsymbol{h}_L$ where $L$ denotes the total number of deep layers. The prediction step is performed by concatenating the outputs from the cross network and the deep network and feeding them

Table 1: The AUC results of nudge CTR prediction in a general case

| Approach | AUC |
|---|---|
| Deep-and-cross model | 0.6421 |
| Caregraph model | 0.7142 |
| Caregraph+TextSim model | 0.7156 |
| Caregraph+TextEmb model | 0.7113 |

to the logit layer. The sigmoid function is used to calculate the final prediction on how likely the user is to engage with the recommended nudge.

## 6.2 TRAINING THE CAREGRAPH MODEL

We trained our knowledge graph embeddings for all nudges, nudge attributes and attribute types (relation types). The embedding sizes $e$ in the knowledge graph embedding are set to 100. We use the pretrained Universal Sentence Encoder from Tensorflow Hub (https://www.tensorflow.org/hub) to extract the text embeddings.

We use Adam to train both of our recommendation models and the deep-and-cross model by optimizing log loss. We apply hyper-parameters search for knowledge graph embedding size, hidden layer units and L2 regularization. The hidden layer in equation 2, 5 and 8 are set to 24 dimensions. For the deep and cross model, the final model architecture is two hidden layers with 240 and 48 dimensions in a deep network and 2 cross layers in the cross network. To evaluate recommendation results with the deep-and-cross model, we use $AUC$ as the evaluation metric.

## 6.3 RESULTS

We evaluate our proposed model with the baseline model in a general case and cold-start situations.

**General case** This objective of this experiment is to compare recommendation results of our proposed models with the baseline model in a overall case. We split the training, validation and testing data into 60:10:30 respectively. We compared AUC from our three proposed models: *CareGraph* (section 4.2), *CareGraph+TextSim* (section 5) and *CareGraph+TextEmb* (section 5) with the deep-and-cross model. The results are shown in Table 1.

From the table 1, all CareGraph approaches outperform a deep-and-cross model. We have tried two different model architectures that integrate text information into our CareGraph model. Our initial hypothesis was that the text embeddings would improve recommendation accuracy because similarity could be computed on more underlying data. However, results showed that this was not the case. AUC did not improve in the text-enabled treatments. A possible reason for this unexpected result is that the kg-embeddings may have already captured enough information to generate high quality predictions. Further studies of model architectures CareGraph with text may be required to prove that this generalizes to all integrations of text similarity to the model.

**Cold start experiments** We simulated cold-start cases by using a leave-one-out approach. Among 155 nudges, one nudge is selected as testing data and keep the the rest nudges for training. We repeated this process for all 155 nudges and compare all AUC with the baseline. In this experiment, we compare our CareGraph model with the baseline model. For the baseline approach, since the deep-and-cross model cannot learn the nudge embedding for new nudge in the training process, we use the prediction results of the most text-based similarity nudge in the training data for a new nudge. In average, our approach has 61.76% AUC and the baseline approach have 58.17% AUC. Our model outperform the baseline model for 96 nudges out of 155 nudges. The distribution of $AUC$ over all nudges shows in Figure 2.

**Cold start analysis** The main objective of using a knowledge graph embedding is to overcome the cold-start and data sparsity problems. A key question in this research is whether knowledge graph embeddings of nudges, nudge attributes, and attribute types capture enough information for new nudge prediction. If we have a new nudge that is different from existing nudges, how that will impact to prediction performance?

In the cold-start experiment, we compare the prediction results with a text-based similarity approach. Text-similarity based approach is heavily depends on existing nudges in the training dataset. If a text message of new nudge has different meaning from the existing nudges, the prediction result can be

| Text similarity to the nearest nudge | | |
| --- | --- | --- |
| approach | $< 0.5$ | $\geq 0.5$ |
| our CareGraph | 0.6176 | 0.6127 |
| deep-and-cross model | 0.5816 | 0.5822 |

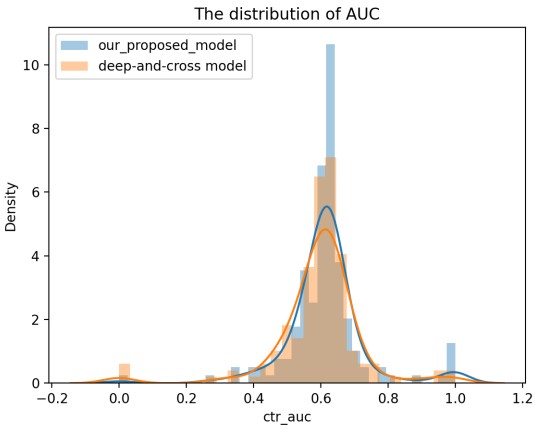
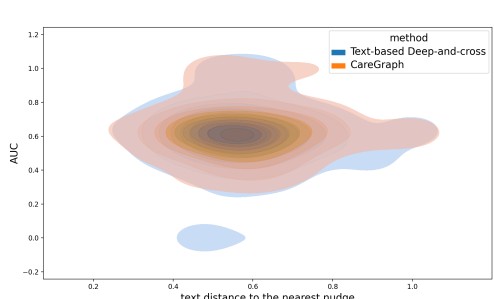

Figure 2: Left: The distributions of $AUC$, and the distribution w.r.t text similarity .

less precise. In our mobile application, it is very likely that we will develop new content. For the knowledge graph embeddings based approach, the embeddings of each nudge preserve connection structures with other entities. Even when we have a new nudge from a new service or program, as long as there are shared attributes with existing nudges, the kg-embedding of new nudge will be not be too far away from existing nudges in the vector space. Also, by adding our nudge attribute embedding $\boldsymbol{n}_{u_i}$ in Equation 5 into the user embedding $e_{u_i}$, this embedding will show how this new nudge is similar to user attribute preferences.

In Figure 2, we compare the AUC distributions of CareGraph and the text-based similarity deep-and-cross approaches over the text distance. The $x$-axis shows the nearest text cosine-similarity of each new nudge to existing nudges. A lower distance means a new nudge content is different from its nearest existing nudge. At the text distances ranging between 0.4 and 0.6, there is a group of nudges that the baseline approach has lower AUC than our proposed approach. Also, if we consider the AUC results of nudges whose nearest text cosine similarity is less than 0.5, our approach improves the AUC by 6.17% and 5.24% for greater than or equal to 0.5. This result shows that we improve AUC more if a new nudge has more different content than existing nudges.

## 7 FUTURE WORK AND CONCLUSION

This research produced several avenues for future work, including analysing different methods to integrate text-based similarity into the model to improve both regular performance and performance under cold start conditions. To conclude, the primary research question in this work asked whether a knowledge graph can be applied to mitigate cold start problems, in the specific task of recommending a finite set of highly structured health nudge messages. To facilitate this we developed a novel knowledge graph based recommender system, which we called *CareGraph*, for predicting health nudges to manage chronic disease on a large digital health platform. Our results confirmed that the knowledge graph does improve predictive accuracy in cold-start situations, showing a 5% improvement over a benchmark model. In addition, a text-similarity based model was evaluated and it was shown that structure in the knowledge graph produced more accurate recommendations than those based on text similarity.

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

# A KNOWLEDGE GRAPH ATTRIBUTES AND EXAMPLE

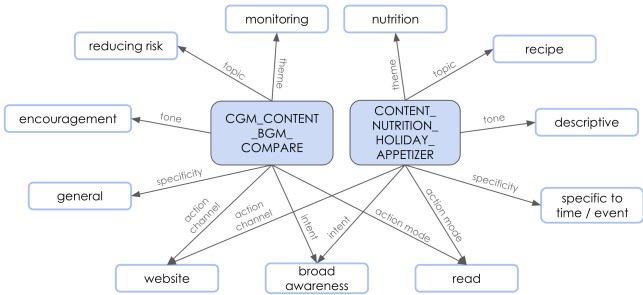

Figure 3: An example of our knowledge graph of nudges

| Attribute Type | Description | Example Value |
|---|---|---|
| Theme | Top diabetes self-management categories such as healthy eating, physical activity, monitoring etc. | Nutrition |
| Topics | Key topics around which nudges are designed, such as Reducing Risk, Education, Event participation etc. | Recipe |
| Action Mode | Attribute to capture the level of effort needed to complete the suggestion action. e.g. Schedule a coaching session, Read content on the mobile application, etc. | Read/View a content |
| Action Channel or Surface | Channel or destination for an required action e.g. redirect to a website, go to the app, redirect to coach scheduling channel. | Website (redirect) |
| Tone | Mood of messages e.g. encouragement, descriptive etc. | Descriptive |
| Intent | Intention categories e.g. Broad awareness, Alert, Suggestion | Broad awareness |
| Specificity | e.g. general, specific to a group of users or occasion. | General |

Table 2: Attribute types, descriptions and example values for the health nudge *"In a dinner rut? Our meal plan is packed with tasty, colorful, and super-healthy recipes, Want it?"*

