# OpenReview forum: "CareGraph: A Graph-based Recommender System for Diabetes Self-Care"
_ICLR.cc/2022/Conference — ICLR 2022 Submitted_

### Official Review · Reviewer_qBZ3 · 2021-10-21

**Correctness:** 3
**Technical Novelty And Significance:** 2
**Empirical Novelty And Significance:** 3
**Recommendation:** 3
**Confidence:** 4

**Main Review:**

Strength:
   1. The application is interesting.
   2. This paper is well organized and easy to follow.
weakness:
   1. The technical novelty of the proposed method is limited.
   2. The experiment is not convincing. The baseline is weak and the experiment lacks an online metric.

significance & novelty.
The proposed application is interesting and useful for diabetes self-care management. But the technical significance is limited. The technique of knowledge graph based recommendations has been investigated in many areas. The technical novelty is limited.

Soundness.
The experiment is not convincing. The paper only compares with DCN that is not a KG model. This paper should compare with other KG models. In addition, in published recommendation system benchmarks, DCN cannot represent the STOA. Moreover, DCN has evolved to DCN-V2. As an offline experiment, only comparing on AUC is not such sound. It is a better way to compare online.





**Summary Of The Paper:**

This paper proposes a graph-based recommender system for diabetes self-care management. The proposed method bases on the knowledge graph embeddings techniques. The proposed method shows better performance compared with two baselines on metric AUC.

**Summary Of The Review:**

This paper proposes a graph-based recommender system for diabetes self-care management, which is based on the knowledge graph embeddings techniques. However, the technical novelty and significance are limited. The reviewer tends to reject the paper.

---

### Official Review · Reviewer_ScTh · 2021-10-31

**Correctness:** 2
**Technical Novelty And Significance:** 1
**Empirical Novelty And Significance:** 1
**Recommendation:** 3
**Confidence:** 4

**Main Review:**

Strengths:
1. The scenario studied in this paper is interesting.
2. This paper is easy to understand.

Weaknesses:
1. The technical contribution of this paper is quite limited. Using knowledge graphs to enhance recommendation is a widely studied problem. Using user preferences to select entity neighbors is also a common technique [1][2]. The knowledge graph embedding method used in this work is also the conventional TransE method. Using textual information is also a common problem in the recommendation field, such as news recommendation [3]. Thus, the distinct contributions of this work are quite limited.
2. Some important details are missing. For example, the construction of the knowledge graph is unclear. The experimental settings such as hyperparameters are not complete.
3. The compared baselines are too weak and limited. There are a large number of methods for knowledge-aware recommendation. The authors should at least include some of them in the comparison.
4. The text embeddings given by the pretrained USE model may not be suitable for recommendation. Why do not include raw textual information and jointly learn the text model with recommendation signals (such as DKN)?

Minors:
Some typos need to be corrected. Just to name a few: "some of the key problems we face is", "individual users", "Zhang et al. (2018) extends.

[1] Wang et al. "Ripplenet: Propagating user preferences on the knowledge graph for recommender systems." Proceedings of the 27th ACM International Conference on Information and Knowledge Management. 2018.
[2] Guo et al. "A survey on knowledge graph-based recommender systems." IEEE Transactions on Knowledge and Data Engineering (2020).
[3] Wu et al. "Personalized News Recommendation: A Survey." arXiv preprint arXiv:2106.08934 (2021).

**Summary Of The Paper:**

This paper introduces a knowledge-graph enhanced recommendation method for healthcare platforms. The authors use user profile information to select entity neighbors for user modeling. Experiments on a nudge CTR prediction dataset show some improvements brought by the proposed method.

**Summary Of The Review:**

This paper suffers from the issues of insufficient contributions, flawed experiments and missing details. Thus, my recommendation is a rejection.

---

### Official Review · Reviewer_r8hu · 2021-11-02

**Correctness:** 2
**Technical Novelty And Significance:** 2
**Empirical Novelty And Significance:** 2
**Recommendation:** 5
**Confidence:** 5

**Main Review:**

Strengths
The use of knowledge graphs in a recommender system is a timely, trend topic of research, especially in the healthcare context.

Weaknesses
It is difficult for me to assess the quality of the work because

(a) the results don't include sufficient recommendations (output) generated from the proposed framework, therefore, it is hard to assess how the proposed approach can be used for a disease or as proposed diabetes care management;
- On page 12, under the caption of Table 2, an example is given. However, it is not clear regarding how it is used as a recommendation. I would suggest including some supporting examples such as how these nudges can be used to manage a disease in the body of the paper, and supplying information and examples regarding how recommendations are assigned to users (i.e. predicted probabilities).

(b) In Section 6.2 under the training the model, hyperparameters/optimization is not described sufficiently (e.g., hyperparameter tuning in deep neural network and choice of optimal settings based on the comparison of results using different hyperparameter settings (e.g., learning rate, batch size)).

(c) lack of validation approaches for the output.
- how did authors validate the knowledge inference (output) (such as correctness or safety since they are recommended to patients  with diabetes)?



**Summary Of The Paper:**

Paper introduces a deep neural network-based recommender that uses the knowledge graph embeddings, called CareGraph to predict health nudges such as content and notifications in a digital health platform to manage a chronic disease. The research question is whether a knowledge graph can be used to mitigate cold start problems to recommend a set of highly structured health nudge messages.

**Summary Of The Review:**

I would suggest a revision for the paper involving a set of results (recommendations) of deep neural network, with clear illustration of input data (ontology of diabetes care). I would also suggest hyperparameter tuning, and comparison of results based on this. Since the recommendations are proposed to manage diabetes care, I would suggest inclusion of a validation approach (e.g., correctness) to make sure that the inferred knowledge can be used as a recommendation and safe for users.

Some minor comments. In general, the relations of knowledge graphs are represented in the form of is-a, part-of, or has-a. In the given example (Fig 3) on page 12 “theme” is stated as a relation, and the ‘nutrition’ (on page 12, Table 2) is its an exemplar value.  Because theme is defined as a relation, so nutrition should not be its instance.  Also, is “encouragement” an instance of a concept (node) or is it a concept?  The defined ontological concepts and their values seem confusing. I think these need to be clarified. These are used in mathematical model of the paper. That would be also nice if development/implementation of the framework platform/software (e.g., python) is specified for readers.

---

### Official Review · Reviewer_zvjx · 2021-11-02

**Correctness:** 2
**Technical Novelty And Significance:** 2
**Empirical Novelty And Significance:** 2
**Recommendation:** 3
**Confidence:** 4

**Main Review:**

In this paper, the authors developed knowledge graph based recommendation approach for recommending healthcare related nudges to mobile users. The key challenge is claimed as the cold start problem in this domain. To this end, they built a knowledge graph where entities are nudges and predefined nudge attribute values, and links are attribute types. Then TransE was employed to learn nudge and attribute embeddings. Built upon the embeddings, the second step method involves a series of transformations for learning user and nudge embeddings, which were used for computing recommendation score through their similarities. Overall, the proposed method is straightforward, which follows commonly used workflow that uses graphs for recommendation. It has several new designs that are specific to the nudge dataset, such as user embedding based on historical nudge clicks, nudge attributes and demographics. However, these designs are sort of incremental and the proposed technique has limited novelty.

The proposed method was claimed for a healthcare application, but it seems the technique is general to nudge recommendation. The paper has few discussion on the specificity of the application domain of healthcare, and how the proposed method was designed to accommodate those domain specific challenges.

In terms of the cold start challenge, from the current paper, it is difficult to understand how could the cold start problem be more severe in this investigated domain than other domains. The paper doesn't highlight and elaborate the challenge. Also, the dataset description has no information about empty users/nudges that are related to the cold start problem.

Because the investigated problem is specific to one domain, the problem description is important and should be clear. However, the current introduction section is obscure, in how nudges are created, what types of data are included in a nudge, whether the set of attribute values are fixed, what is the rule to set attributes and attribute values. The figure and table in the appendix are important (but doesn't answer all the questions), and are better be moved to the main text.

As for the technical design, there are some assumptions and simplifications that may not be reasonable. For example, the assumption that users are interested in new nudges that are similar to their historical nudges may not be true, as users may not be interested in duplicate messages. The authors also mentioned the importance of diversity, but it appears the proposed method didn't have explicit designs to encourage diversity of recommended nudges. Also, the average embedding in Eq 2 may be simplified as the temporal information of the historical nudges has not been taken into account. Similarly, the zero embedding in Eq 2 for users without history, and the random selection of nudge attributes for these users are sort of arbitrary. The ratio of such users are unclear from the paper, and the impacts these processing steps for them haven't been discussed. Finally, some designs are not well justified. In Eq 5, it is not clear why the transformation step was included.

The authors also considered to integrate the texts of nudges in the proposed model, but the experimental results didn't demonstrate the effectiveness of the texts. The reason could be from the complexity of the data, or the simplification of methods to incorporate the text embeddings.

The experiments remain to be improved. First, only a private dataset was used, thus it is hard to judge how general the proposed method is. In the description of the dataset, important information such as the number of nudges, attributes, attribute values, historical nudges per users, are absent. For comparison, there is only one compared method, but it is unclear why that method was selected as the sole baseline. The evaluation metric only has AUC, other commonly used metrics in recommendation could also be used to provide multiple perspectives of evaluation. In the cold start setting, it is unclear how a new nudge's embedding was learned. Since the knowledge graph was trained to learn all embeddings, does the proposed method re-train the knowledge graph once a new nudge is added?

Finally, the presentation of the paper could be improved as there are grammar errors and typos that make reading not easy.

**Summary Of The Paper:**

This paper investigates the nudge recommendation problem in a mobile healthcare domain, and introduces a knowledge graph based recommendation approach. The proposed method builds a nudge-attribute knowledge graph and uses TransE to infer nudge and attribute embeddings, which are used in a downstream step that leverages embedding similarity for recommendation.

**Summary Of The Review:**

The paper provides a generally reasonable framework to handle the nudge recommendation problem, but the proposed method is incremental upon existing techniques. It could be seen as a good exploration of applications of those techniques with proper integration. The paper also remains to be improved in its technical details, presentation and experiments.

---

### Decision · Program_Chairs · 2022-01-20

**Decision:**

Reject

**Comment:**

The paper introduces the CareGraph, a knowledge graph based recommendation approach.
CareGraph is a deep neural network-based recommender that can be used a mobile healthcare platform
for nudge recommendation. The main motivation is to use the knowledge graph to
mitigate cold start problems when recommending nudge messages.

The papers' main strength is the topic of interest. Research on recommending systems in the healthcare context is of great interest.
However, the reviews raised concerns that outweigh the strengths.
The majority reviewers agree that the work is not ready for publication.
Main concerns focus on weak experimental section and lack of technical details.

I recommend the authors to incorporate all the reviewers' comments and make a
stronger submission to a future conference!